# A Mitigation Method for Optical-Turbulence-Induced Errors and Optimal Target Design in Vision-Based Displacement Measurement

**DOI:** 10.3390/s23041884

**Published:** 2023-02-08

**Authors:** Xingyu Huang, Wujiao Dai, Yunsheng Zhang, Lei Xing, Yichao Ye

**Affiliations:** School of Geosciences and Info-Physics, Central South University, Changsha 410083, China

**Keywords:** computer vision, displacement measurement, optical-turbulence error, optimal target

## Abstract

Computer vision-based displacement measurement techniques are increasingly used for structural health monitoring. However, the vision sensors employed are easily affected by optical turbulence when capturing images of the structure, resulting in displacement measurement errors that significantly reduce the accuracy required in engineering applications. Hence, this paper develops a multi-measurement point method to address this problem by mitigating optical-turbulence errors with spatial randomness. Then, the effectiveness of the proposed method in mitigating optical-turbulence errors is verified by static target experiments, in which the RMSE correction rate can reach up to 82%. Meanwhile, the effects of target size and the number of measurement points on the proposed method are evaluated, and the optimal target design criteria are proposed to improve our method’s performance in mitigating optical-turbulence errors under different measurement conditions. Additionally, extensive dynamic target experiments reveal that the proposed method achieves an RMSE correction rate of 69% after mitigating the optical-turbulence error. The experimental results demonstrate that the proposed method improves the visual displacement measurement accuracy and retains the detailed information of the displacement measurement results.

## 1. Introduction

With the aging of the engineering structures, environmental changes, and accidents, engineering structures will suffer different degrees of damage, leading to material and property losses and potentially endangering human lives. To detect damage and harmful deformation of engineering structures timely and guarantee their long-term safe use, evaluating and maintaining the structures in time is mandatory [1,2]. Specifically, displacement is an important structural condition assessment and performance evaluation index [3,4], with computer vision-based displacement measurement systems recently applied for structural health monitoring (SHM) presenting great success. Visual displacement measurement systems employ non-contact sensors to measure the target and obtain the structure’s displacement, realizing long-distance, non-contact, high-precision, low-cost, multi-point, high-frequency, and real-time monitoring, presenting significant advantages over the displacement measurement method relying on traditional sensors [5,6,7,8]. However, vision sensors are easily affected by optical turbulence during vision measurements [9,10]. Indeed, as the atmospheric temperature gradient changes continuously, the air density changes irregularly, and the atmospheric refractive index becomes inhomogeneous, resulting in the complex optical phenomenon of optical turbulence. This is because when light travels in the atmosphere, the light from the same source is affected by optical turbulence and will reach the camera image plane through a random path, forcing the target image captured by the camera to suffer from blurring, dithering, offset, and random noise [11,12]. Therefore, the displacement measurement error due to optical turbulence will be mixed with the real displacement of the structure. If the optical-turbulence error cannot be mitigated, the demand for high-precision structural displacement measurement cannot be satisfied.

Current research mitigating the optical turbulence effects involves four categories. The first one is image post-processing to restore images affected by optical turbulence. For example, “lucky-region” fusion (LRF) approach [13], lucky image approach [14], region-level fusion based on dual tree complex wavelet transform [15], B-spline-based nonrigid registration [16], space-variant blind-deconvolution [17], optical-flow based super resolution [18], derivative compressed sensing [19], neural networks [20], etc. Although such methods effectively mitigate image geometric distortion and blurring, a significant amount of detailed information in the image is lost. Thus, it will lose the structure’s real displacement information if it is applied for computer-based vision displacement measurement and reduce its accuracy. The second category relies on a stable background and uses fixed background objects as a reference, e.g., buildings, mountains, and bridge piers, of the image presenting the monitoring structure, thus mitigating the measurement error induced by optical turbulence in visual displacement measurements [21,22,23]. However, such a strategy is difficult to implement in practical engineering applications, as the optical turbulence increases with the measurement distance [24], making it difficult to find a stationary object in or near the same plane of the structure as a reference background. The third category mitigates optical-turbulence errors by capturing images of different spatial locations simultaneously and then filters them, exploiting the spatial randomness of the optical-turbulence errors. This category is divided into multi-point and multi-view monitoring. The former selects multi-measurement points of the structure to mitigate the effect of optical-turbulence errors by spatially filtering the acquired multi-point displacements [23]. The latter simultaneously measures the structure with multiple cameras installed at different angles to mitigate optical-turbulence errors [25]. This method does not reduce the temporal resolution of the visual displacement measurement results but preserves the real displacement information of the structure to a large extent. However, this category has several limitations, such as multi-point monitoring being challenging to apply on structures with less natural texture and imposing high operating costs and low camera synchronization accuracy. The fourth category is based on frequency domain filtering and time domain filtering methods. When the frequency of the errors induced by optical turbulence is significantly different from the structure frequency, frequency domain filtering can be used to mitigate the optical-turbulence errors. In practical projects, however, the frequency of the optical-turbulence errors is often mixed with the frequency of the structural [23], and the effect of using frequency domain filtering to mitigate the optical-turbulence errors is poor. Time domain filtering, such as sliding-average method, uses the temporal diversity of optical turbulence to mitigate the high frequency, random optical-turbulence errors, but time domain filtering will reduce the temporal resolution of the displacement data and lose the real displacement information of the structure.

So far, the effect of optical turbulence on vision-based measurements has rarely been mentioned, but optical-turbulence errors can reach 50 mm [26], which shows the importance and necessity of this study. Spurred by the existing research and problems, this paper proposes a method to mitigate the optical-turbulence error using multi-measurement points on the target based on the spatial randomness of the optical-turbulence error. Experimental evaluation of static target displacement measurements reveals that our method’s accuracy and efficiency in mitigating optical-turbulence errors are closely related to the target size and the number of measurement points. Accordingly, we propose the design criteria of the optimal target, providing a reference standard for the parameter selection for optical-turbulence error mitigation in practical engineering. Moreover, we verify the proposed method’s accuracy and reliability to mitigate the optical-turbulence errors.

The remainder of the paper is organized as follows. Section 2 describes the proposed optical-turbulence error mitigation and displacement calculation method. Section 3 experimentally validates our method, including static and dynamic target experiments. Finally, Section 4 concludes that this paper proposes some future research directions.

## 2. Methods

Studies investigating the effect of optical turbulence on vision-based displacement measurements reveal that optical-turbulence errors are random in the temporal and spatial domains [27]. Therefore, using the displacement measurement results of multi-measurement points at different locations on the target for spatial filtering mitigates optical-turbulence errors. To exploit the spatially random character of the optical- turbulence error, we employ as a target a checkerboard with multi-measurement points (Step 1 in Figure 1). Based on the above analysis, the spatial distribution and magnitude of displacement measurement errors of the multi-measurement points on the checkerboard, under the influence of optical turbulence, are spatially random. However, since the target is rigid and the real displacements of the multi-measurement points on the target are isotropic, this paper proposes an optical-turbulence error mitigation method using the average displacement of the multi-measurement points on the target. This strategy obtains more accurate displacement measurements of the structure. Figure 1 illustrates the proposed architecture, comprising five steps.

Step 1: Install the visual displacement measurement system to acquire the target images on the structure continuously. Then use the first image captured by the visual displacement measurement system as the reference frame in the subsequent displacement calculation.

Step 2: Detect and locate the image coordinates of the multi-measurement points on the target image. This paper employs a Radon transform-based method to enhance the saliency of the center region of the cross-shaped corner points [28] and accurately localize the results of the measurement points on the checkerboard.

Step 3: Calculate the displacement value $D_{k}^{i}$ of each measurement point in the subsequent frames of the target image and on the corresponding reference frame based on the positioning result.
(1)$$D_{k}^{i}=Y_{k}^{i}-Y_{0}^{i}\;(k=1\ldots ,\;N;i=1\ldots \;n)$$
where $Y_{k}^{i}$ and $Y_{0}^{i}$ are the coordinates of the ith measurement point in the subsequent frame and the reference frame, n is the total number of measurement points, k is the number of current image frames, and N is the total number of image frames.

Step 4: The displacements of all measured points on the same frame of the target are averaged to obtain the displacement $D_{k}^{c}$ for each image frame after mitigating the optical-turbulence error.
(2)$$D_{k}^{c}=\frac{\sum_{i=1}^{i}D_{k}^{i}}{n}\;(k=1\ldots \;N;i=1\ldots \;n)$$

Step 5: The scale factor S is calculated based on the target’s pixel size $L_{\mathrm{image}}$ and the physical size $L_{\mathrm{physical}}$. Moreover, the target’s pixel displacement is converted to the real physical displacement $D_{k}^{r}$.
(3)$$S=\frac{L_{\mathrm{physical}}}{L_{\mathrm{image}}}$$
(4)$$D_{k}^{r}=S\times D_{k}^{c}\;(k=1\ldots \;N)$$

## 3. Experiments and Analysis

We conduct displacement measurement experiments involving static and dynamic targets to verify the proposed method’s effectiveness regarding the target’s multi-measurement points in mitigating the optical-turbulence error. The employed computer vision-based displacement measurement system involves an industrial camera, lens, computer, tripod, and data cable. Among these components, the industrial camera (MV-SUA230GM) with 75 mm telephoto lens is used, while the computer is equipped with an Intel(R) Core (TM) i7-9750H CPU @ 2.60 GHz with 16.0 GB RAM. More detail parameters are shown in Table 1.

### 3.1. Static Target Experiment

The computer vision-based displacement measurement system is stably installed in a closed laboratory with a measurement distance of 5 m between the camera and the target. Moreover, a heating furnace (up to 300 °C) placed at the mid-position simulates the optical turbulence. The camera and the target always remain stationary, so the true displacement of the target is zero, and the displacement measurement error is only induced by optical turbulence. When the heating furnace temperature (300 °C) reaches a steady state, the target image is captured at a camera sampling rate of 90 Fps and a sampling time of 10 s. Then, the coordinates of the measurement points on the target are detected, their displacements are calculated, and the optical turbulence effect on the visual displacement measurements is mitigated using the proposed method. The experimental setup is depicted in Figure 2.

Eight target sets with a different number of measurement points at different sizes are used to evaluate the effect on the performance of the optical-turbulence error mitigation method. The target’s side length is 55, 75, 95, 115, 135, 145, 165, and 185 mm, the corresponding image sizes are 285, 380, 475, 570, 665, 760, 855, and 950 pixels, and the number of measurement points on the target is 1, 4, 16, 36, 64, 100, 144, 196, 256, 324, 400, and 484, respectively. Since the targets must be changed frequently, we employ a laptop screen to display the different measurement targets, avoiding unnecessary errors. Figure 3 illustrates a set of target images of different measurement point targets affected by optical turbulence, as acquired by the computer vision-based displacement measurement system. The measurement points at different locations on the target appear, such as irregular geometric distortion, blurring, and jittering. Precisely, the number of measurement points is large, and the measurement points with a large distortion will have large errors or are difficult to identify when detected and positioned, leading to displacement measurement errors. Thus, the influence of optical turbulence on the vision-based displacement measurement cannot be neglected.

The target images acquired in the experiment are processed by the developed method to obtain the displacement measurement results after mitigating the optical-turbulence error. For example, Figure 4 illustrates the optical displacement measurements in the vertical direction before and after mitigating the optical-turbulence error for measurement points of 1, 36, 256, and 484 at a side length of 570 pixels. To visualize our method’s effectiveness, the displacement measurement results before mitigating the optical-turbulence error are presented as gray curves (only the displacement measurement error of one measurement point is shown), and the displacement measurement results after mitigating the optical-turbulence error are shown as blue curves. Figure 4 reveals that the displacement measurement results are affected by atmospheric turbulence presenting many random fluctuation errors, seriously affecting the accuracy of the visual displacement measurement. Moreover, the displacement measurement results are affected by optical turbulence having many random fluctuation errors, significantly affecting the accuracy of the vision-based displacement measurement. In this case, the displacement measurement before and after mitigation is the same when the number of measurement points is one. As the number of measurement points on the target increases, the displacement measurement error significantly reduces when using the proposed method to mitigate the optical-turbulence error.

Furthermore, we calculate the RMSE of the displacement measurements for different targets before and after the optical-turbulence error mitigation, with Figure 5 displaying the results. It should be noted that the RMSE before mitigating the optical-turbulence error is the average RMSE of the displacement measurements at all measurement points on the target. Additionally, the closer the RMSE is to zero, the more effective our method is in mitigating the optical-turbulence errors. Figure 6 presents the RMSE correction rate, while Figure 5 reveals that the RMSE of the different targets displacement measurement results range from 0.531 to 0.940 pixels before the optical-turbulence error is mitigated. The results of optical-turbulence error mitigation vary among different targets after using the proposed method, and the RMSE correction rate is up to 82%.

To illustrate the effect of the target size and measurement points on the proposed method’s effectiveness in mitigating optical-turbulence errors, we calculate the correlation coefficients for the optical-turbulence errors between the measurement points using the Pearson correlation coefficient method [29] (Figure 7). The closer the absolute value of the correlation coefficient is to one, the stronger the correlation between the two measurement points. In addition, Figure 8 presents statistics of the execution time for the optical-turbulence error mitigation using different targets.

Combining the above results, the RMSE of the displacement measurement results after mitigating the optical-turbulence error is not significantly reduced when the target size is 285 and 380 pixels. Analyzing Figure 7 reveals that the correlation coefficients for targets with 285 and 380 pixel sizes are higher than the larger target size, and thus using the proposed method is less effective in mitigating the optical-turbulence error. As the target size increases, the RMSE of the displacement measurement results after mitigating the optical-turbulence error tends to decrease, and there is a certain improvement in the effect of mitigating the optical-turbulence error using the proposed method. The correlation coefficient shows that the correlation decreases as the target size increases, so the measurement points at different locations on the target can be fully utilized to mitigate the optical-turbulence error. However, the RMSE of the displacement measurement results gradually stabilized after the target size increased to nearly 665 pixels to mitigate optical-turbulence errors and did not continue to decrease significantly with the increase of the target size. By analyzing the correlation coefficients, as the target size keeps increasing, the correlation coefficients keep converging to zero, and our method’s effectiveness in mitigating the optical-turbulence errors is no longer significantly improved. In addition, when the target size keeps increasing, Figure 8 reflects that its execution time also increases. Considering the installation of the target in practical engineering applications, oversized targets will greatly reduce the practicality of the proposed method and add additional usage and computational costs. Therefore, the target size should be controlled.

When one measurement point is used on the target, randomness is not employed to mitigate the optical-turbulence error, and the RMSE reflects the displacement measurement results without mitigating the optical-turbulence error. The displacement measurement RMSE after mitigating the optical-turbulence errors decreases as the number of measurement points increases, and the effectiveness of mitigating optical-turbulence errors by averaging the displacement of measurement points on the target increases. As the number of measurement points increases to about sixteen, the RMSE no longer decreases significantly and gradually plateaus. According to the correlation coefficients, it is known that the correlation coefficients still converge to one as the number of measurement points increases, and the strong correlation limits the proposed method’s effectiveness in mitigating optical-turbulence errors. Moreover, an excessive number of measurement points inevitably reduces the target image’s effective pixels from being detected and localized, making the identification and localization of the measurement points more challenging, and impacting our method’s effectiveness in mitigating optical-turbulence errors. Similarly, Figure 8 reflects that, as the number of measurement points increases, the execution time increases. Therefore, the number of measurement points must be controlled.

In summary, we suggest the following optimal target design criteria to ensure the practicality and effectiveness of the optical-turbulence error mitigation method for engineering applications:Our method’s effectiveness in mitigating optical-turbulence errors is poor when the target image is small, or the number of measurement points is small. Therefore, the target’s image size should be more than 380 pixels, and more than one measurement point should be exploited.When the target image is large, or the number of measurement points is large, using the developed scheme to mitigate the optical-turbulence error is costly to use and calculate. Thus, the difficulty in installing the target on the structure should be considered when designing the target’s size. Therefore, the target must be as small as possible to meet the accuracy requirements of engineering applications, and the minimum number of measurement points should be chosen on this basis.Determine the optimal target image size according to the demand of measurement accuracy, then select the appropriate camera and lens to make the optimal target based on the measurement distance (Equation (5)). When the measurement distance is longer, a camera with smaller image elements and a longer focal length lens should be selected to adjust the target’s actual size, ensuring that the target is smoothly installed on the structure.
(5)$$D=d\cdot \frac{L}{f}d_{\mathrm{pixel}}$$
where D (mm) and d (pixel) are the target’s actual physical size and pixel size, respectively, L (m) is the measurement distance, f (mm) is the focal length of the camera lens, and $d_{\mathrm{pixel}}$ (μm) is the camera image element size.

For example, the RMSE accuracy requirement of displacement measurement in engineering applications is 0.3 pixels. In that case, we choose a target with an image size of 665 pixels for visual displacement measurement, and the parameter combination of the optimal target is reported in Table 2.

Figure 9 illustrates the static target’s power density (PSD) plots before and after mitigating the optical-turbulence error (on the figure’s left and right sides, respectively) to analyze the corresponding spectral changes. By analyzing the frequency spectrum before and after mitigating the optical-turbulence error, we find that the frequency distribution of the displacement measurement results before mitigating the optical-turbulence error has a wide range, which is consistent with the frequency distribution characteristics of random signals. Moreover, the different signals are mixed, proving why it is difficult to filter them in the frequency domain to mitigate the optical-turbulence errors. However, when applying the proposed method, the frequency distribution of the displacement measurement results ranges within 0~10 Hz, reflecting the frequency distribution of the static target displacement measurement results.

### 3.2. Dynamic Target Experiment

Figure 10 depicts the experimental setup used to verify the dynamic target dis-placement measurement. In order to make the accuracy of this displacement measurement system higher than 0.3 pixels, according to the optimal combination of the target parameters in Table 2 obtained from the static experimental analysis, a 260 mm × 260 mm target was made when the measurement distance was 5 m, with 16 measurement points on the target. The target was affixed to a metal bracket, and a computer-based visual-based displacement measurement system was installed 5 m from the target, with a simulated optical turbulence heating furnace (300 °C) placed between the target and the camera. In addition, a camera was placed between the heating furnace and the target to simultaneously measure the real displacement of the target without being influenced by the optical turbulence, which was used as a reference for the dynamic target displacement measurement results. The metal bracket produced high-frequency vibration in the horizontal direction during the experiment by knocking. The sampling rate of both cameras was set to 70 Fps and the sampling time was 10 s.

The displacement measurements of the target in the horizontal direction before and after mitigating the optical-turbulence error are presented in Figure 11. To demonstrate the proposed method’s effectiveness in mitigating the optical-turbulence error in high-frequency dynamic target visual displacement measurements, we present only the displacement measurement set with the smallest error before and after the optical-turbulence error mitigation. Figure 11 highlights that the optical-turbulence error is mixed with the high-frequency motion of the target before the optical-turbulence error is mitigated and the true displacement of the target cannot be obtained if the optical-turbulence error is not mitigated. The average displacement measurement RMSE before the optical-turbulence error mitigation is 0.866 pixels, and after the optical-turbulence error mitigation, it is 0.269 pixels (correction rate of 69%).

Meanwhile, the displacement measurement PSD before mitigating the optical-turbulence error is illustrated in Figure 12, and the frequency of the target is identified from the PSD curve using the peak picking technique [30]. Furthermore, Figure 12a illustrates the PSD of the displacement measurements obtained by the reference camera when unaffected by optical turbulence. Thus the true vibration frequency of the target is 15.12 Hz. Figure 12b illustrates the PSD of the displacement measurement error after being affected by optical turbulence, which is similar to the static displacement experimental results, where the frequency of the optical-turbulence error and the frequency of the target are mixed together. In Figure 12c, after mitigating the optical-turbulence error using the proposed method, the true frequency of the target is clearly reflected and matches well with the results obtained from the reference camera. Nevertheless, when using both frequency and time domain filtering to mitigate the optical-turbulence error, the true displacement frequency of the dynamic target cannot be obtained. The result in Figure 12d reveals that the frequency domain filtering incorrectly retains the frequency of the optical-turbulence error mixed with the true frequency of the target. Figure 12e,f reveals that the time domain filtering scheme induces excessive smoothing, and it becomes more serious as the sliding window increases.

In addition, to verify the proposed method’s effectiveness in mitigating the optical-turbulence error, the visual displacement measurement results are processed with typical time-domain and frequency-domain filtering (Figure 13). The time domain filtering uses a sliding window to average the temporal displacement results, and the frequency domain filtering employs a bandpass filter to process the frequency domain data. The cutoff frequency of the bandpass filter is set from 12 to 18 Hz according to the analysis of Figure 12a. In Figure 13a, after using the proposed method to mitigate the optical-turbulence error, the results reflect the target’s dynamic displacement and match better with the displacement results measured by the reference camera. However, the time domain and frequency domain filtering schemes did not have the desired effect on obtaining the target’s true displacement but lost some significant displacement information. The result of frequency domain filtering is illustrated in Figure 13b, and it can be clearly seen that the displacement result mitigated by the frequency domain filtering incorrectly retains some errors induced by optical turbulence, which has a large deviation from the reference value. Figure 13c–f illustrates the results of the time domain filtering scheme. When the sliding window is small, as shown in Figure 13c, it is unable to filter displacement errors of large magnitude. When the sliding window is large, the time domain filtering schemes lead to excessive smoothing with a large deviation from the actual displacement acquired by the reference camera.

Figure 14 illustrates the calculated RMSE and the maximum error of the experimental results. The maximum RMSE and error are 0.269 pixels, and 1.520 pixels, respectively, which are 49% and 45% corrected compared to the worst results, further proving that time and frequency domain filtering cannot handle high-frequency displacement measurement data.

## 4. Conclusions

Optical turbulence is an important error source in vision-based displacement measurements. Thus, this paper employs multi-measurement point target to mitigate the optical-turbulence error based on the error’s spatial randomness induced by optical turbulence. Meanwhile, we focus on the experimental evaluation of the effect of the target size and the number of measurement points on the method to mitigate the optical-turbulence error, so as to propose the optimal target design criteria. From this study, we conclude the following:(1)The static experiments capturing target images of different sizes and measurement points validate the effectiveness of the proposed method. Among them, the RMSE correction rate is up to 82%. Furthermore, the effectiveness of targets of different sizes and measurement points in mitigating optical-turbulence error is evaluated. According to the results, the optimal target design criteria for mitigating optical-turbulence error are proposed to ensure successful application of the method in different practical projects. Moreover, the parameter combination of the optimal target to mitigate the optical-turbulence error is based on the measurement distance to select the appropriate camera, lens and make the optimal target, as shown in Equation (5).(2)The optimal target designed according to the criteria is used in dynamic target experiments, and the proposed method mitigates the optical-turbulence error. The results demonstrate that the proposed method effectively mitigates the optical-turbulence error in vision-based displacement measurements and improves the optical displacement measurements’ accuracy, and its RMSE correction rate can reach 69%. Compared with frequency domain filtering and time domain filtering, the proposed method can retain the real displacement information of the structure and effectively process the signals mixed in frequency to obtain the true frequency.

This work also has some limits. The effectiveness of the proposed method depends on the optimal target. Complex outdoor environments such as light, fog, rain or shadow may affect the mitigation effect of optical-turbulence error. Therefore, the effectiveness of the proposed method in mitigating the optical-turbulence error can be further validated in subsequent engineering applications.

## Figures and Tables

**Figure 1 sensors-23-01884-f001:**
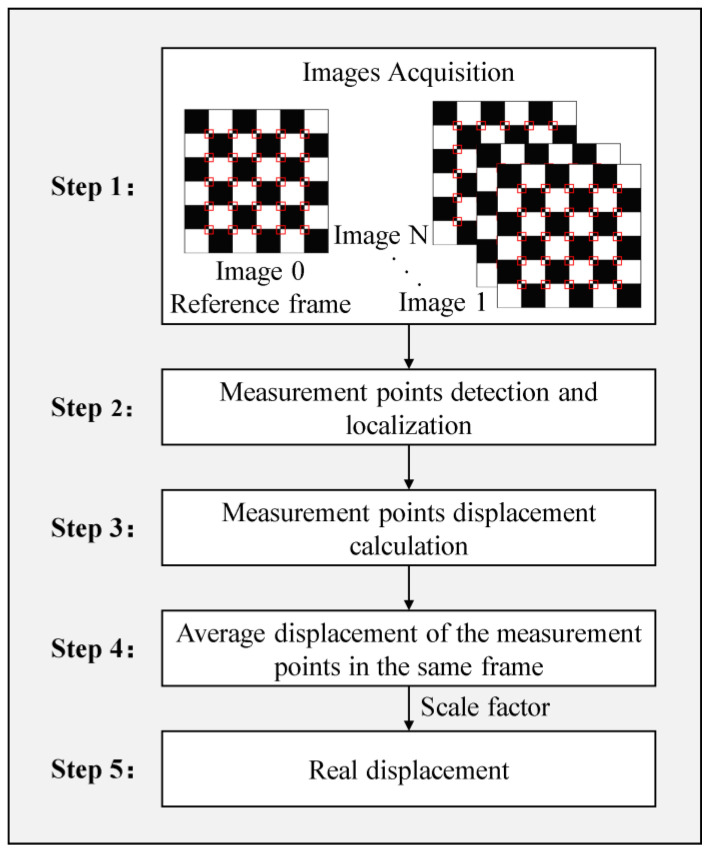
Workflow of the proposed method.

**Figure 2 sensors-23-01884-f002:**
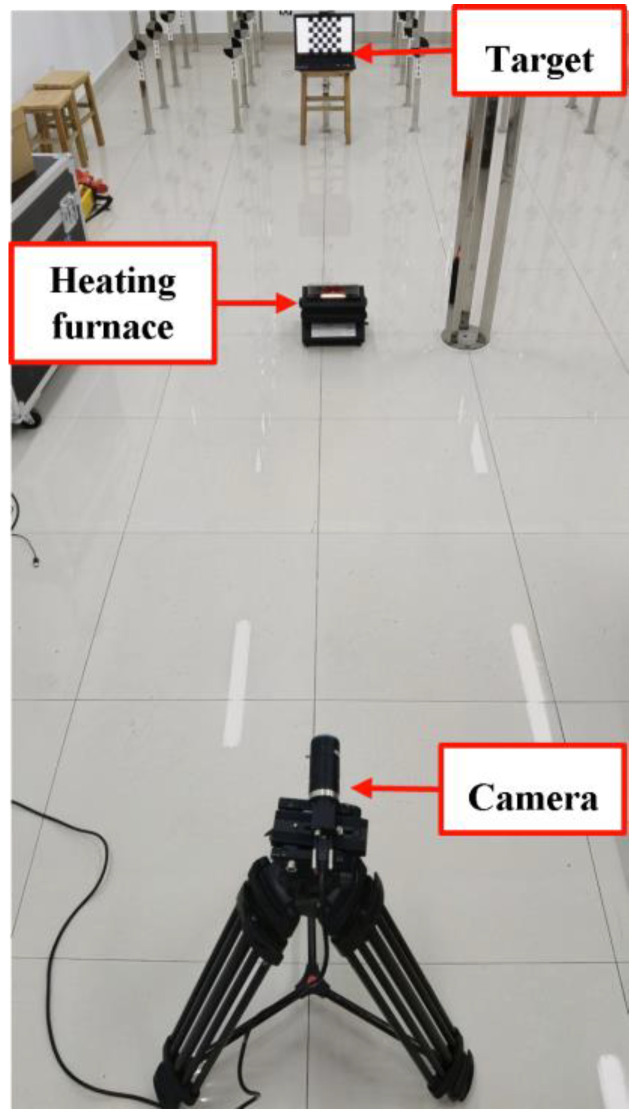
Static target displacement measurement experimental setup.

**Figure 3 sensors-23-01884-f003:**
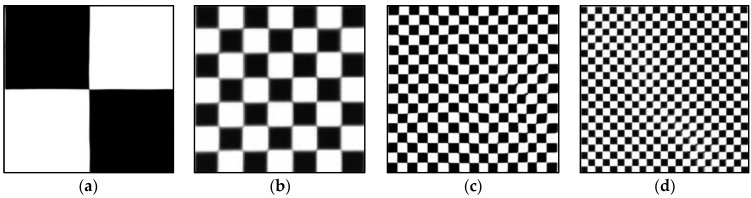
Target images under various numbers of measurement points affected by optical turbulence for a target side length of 285 pixels, Number of measurement points (**a**) 1, (**b**) 36, (**c**) 256, and (**d**) 484.

**Figure 4 sensors-23-01884-f004:**
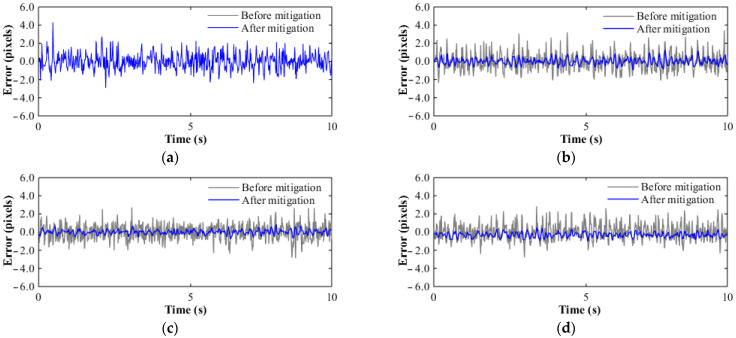
Displacement measurement results before and after optical-turbulence error mitigation. Number of measurement points (**a**) 1, (**b**) 36, (**c**) 256, (**d**) 484.

**Figure 5 sensors-23-01884-f005:**
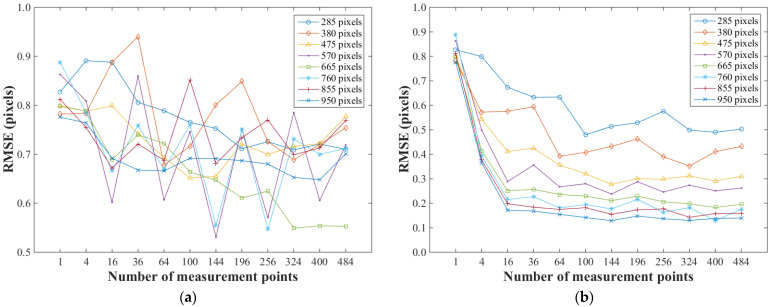
RMSE of the displacement measurement results before and after the optical-turbulence error mitigation for different targets (**a**) before mitigation and (**b**) after mitigation.

**Figure 6 sensors-23-01884-f006:**
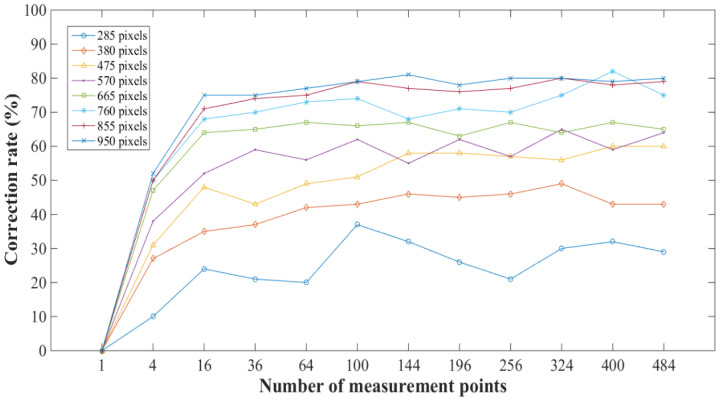
RMSE correction rates of displacement measurement results after optical-turbulence error mitigation for different targets.

**Figure 7 sensors-23-01884-f007:**
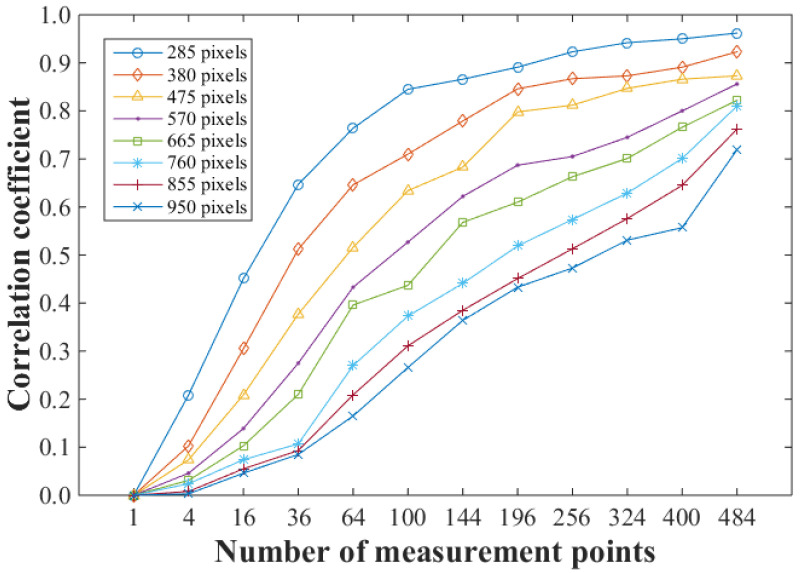
Correlation coefficients of optical-turbulence errors between measurement points on different targets.

**Figure 8 sensors-23-01884-f008:**
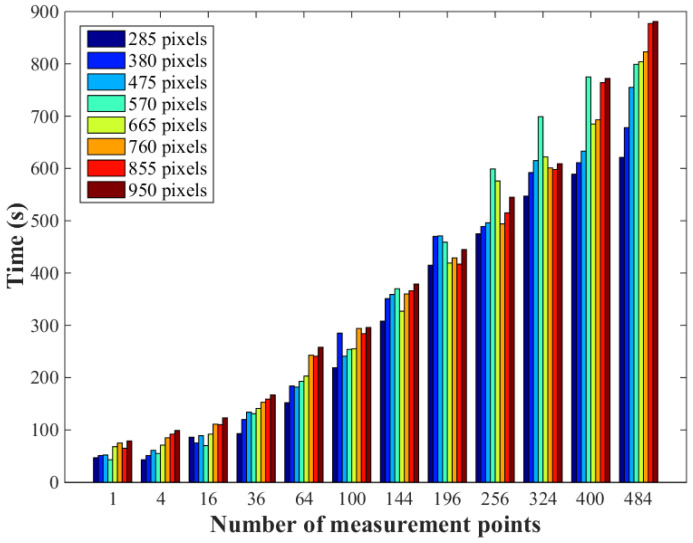
Execution time of optical-turbulence error mitigation for different targets.

**Figure 9 sensors-23-01884-f009:**
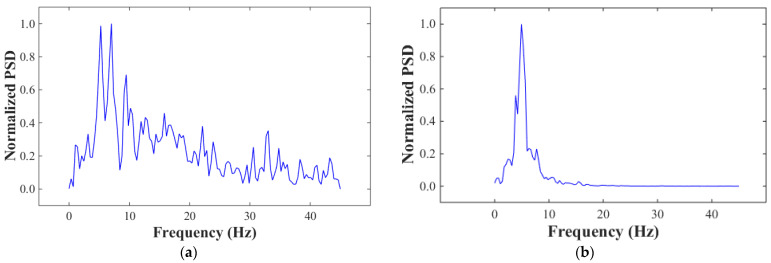
Power spectral density before and after mitigation of optical turbulence errors: (**a**) Before mitigation; (**b**) After mitigation.

**Figure 10 sensors-23-01884-f010:**
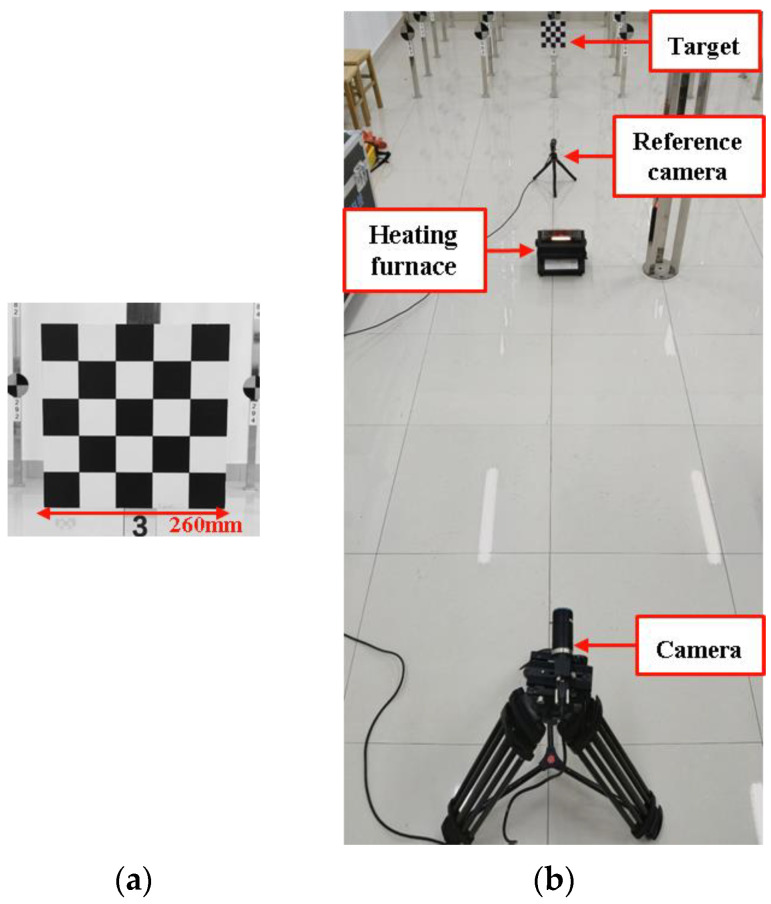
Dynamic target experimental setup: (**a**) Optimal target with a side length of 260 mm; (**b**) Experimental setup.

**Figure 11 sensors-23-01884-f011:**
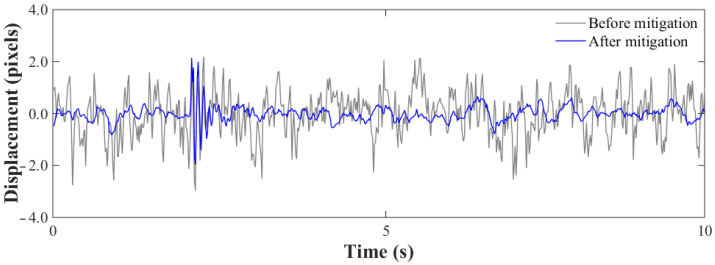
Displacement measurement results before and after optical-turbulence error mitigation.

**Figure 12 sensors-23-01884-f012:**
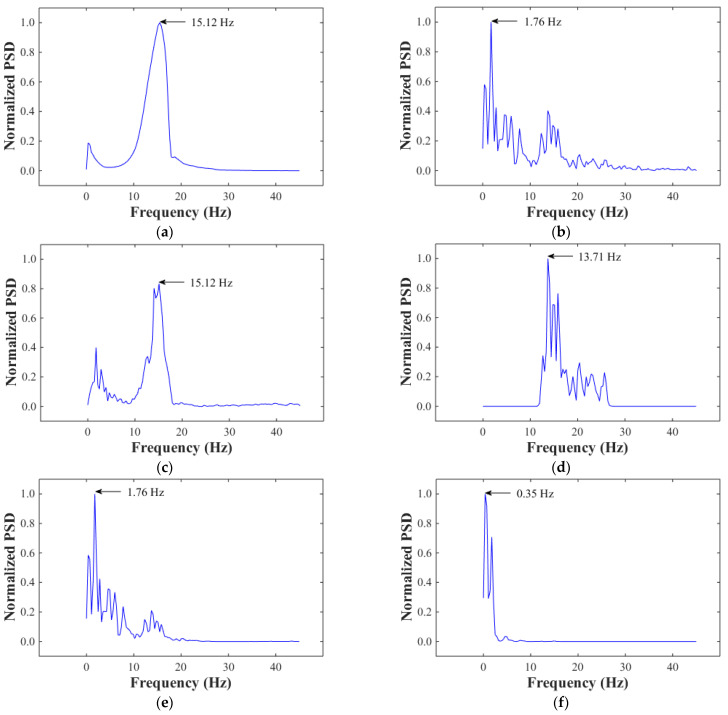
Power spectral density (PSD) before and after optical-turbulence error mitigation: (**a**) Reference; (**b**) Before mitigation; (**c**) After mitigation by the proposed method; (**d**) After mitigation by frequency domain filtering (bandpass filter: 12~18 Hz); (**e**) After time domain filtering mitigation (moving average: window size = 3); (**f**) After time domain filtering mitigation (moving average: window size = 27).

**Figure 13 sensors-23-01884-f013:**
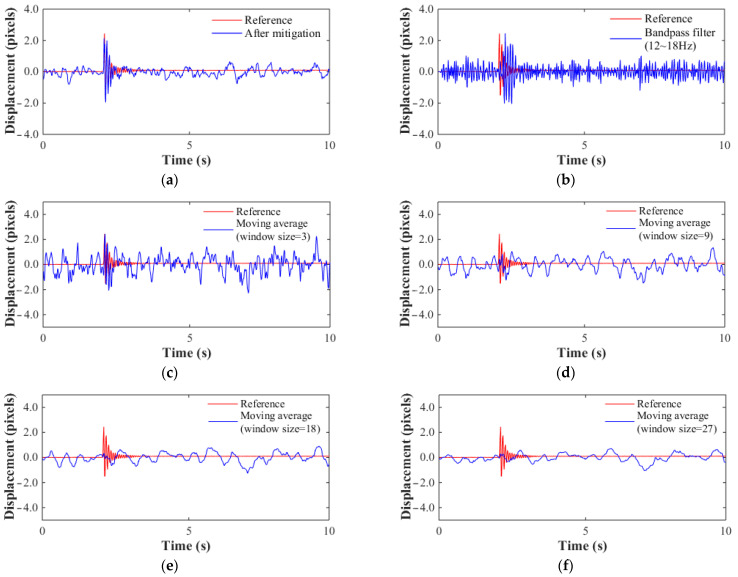
Experimental results, comparison results of the reference camera using (**a**) the proposed method, (**b**) bandpass filter (12~18 Hz), (**c**) moving average (window size = 3), (**d**) moving average (window size = 9), (**e**) moving average (window size = 18), and (**f**) moving average (window size = 27).

**Figure 14 sensors-23-01884-f014:**
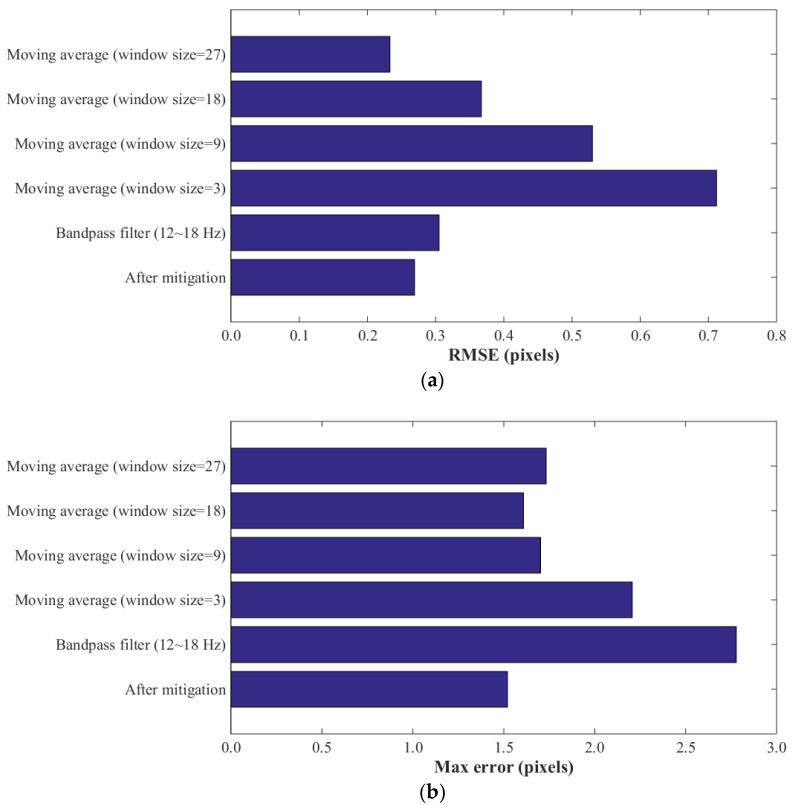
Statistical results of optical-turbulence error mitigation by different methods (**a**) RMSE and (**b**) maximum error.

**Table 1 sensors-23-01884-t001:** The parameters of the displacement measurement system.

Components	Parameters
Camera	Camera size: 29 mm × 29 mm × 32.7 mm Frame rate: 165 FPS Sensor type: CMOS Pixel size: 5.86 μm Maximum resolution: 1920 × 1200 Interface: USB 3.0
Camera lens	Focal length: 75 mm Aperture: F = 2.8 Minimum focus distance: 0.8 m
Computer	Intel(R) Core(TM) i7-9750H CPU @ 2.60 GHz 2.59 GHz RAM: 16.0 GB

**Table 2 sensors-23-01884-t002:** Recommended parameter combinations.

d_pixel_ (μm)	f (mm)	L (m)	D (mm)	d_pixel_ (μm)	f (mm)	L (m)	D (mm)
3.45	35	2.5	164	3.45	120	10	191
4.00	35	2.5	190	4.00	120	10	222
5.86	35	2.5	278	5.86	120	10	325
3.45	75	5	153	3.45	135	15	255
4.00	75	5	177	4.00	135	15	296
5.86	75	5	260	5.86	135	15	433
3.45	100	7.5	172	3.45	165	20	278
4.00	100	7.5	200	4.00	165	20	322
5.86	100	7.5	292	5.86	165	20	472

## Data Availability

Not applicable.

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
