# Peer review of "A Mitigation Method for Optical-Turbulence-Induced Errors and Optimal Target Design in Vision-Based Displacement Measurement"

_sensors, 2023, doi:10.3390/s23041884_

Round 1

Reviewer 1 Report

This paper develops a multi-measurement point method to address this problem by mitigating optical-turbulence errors with spatial randomness. Specially, the effects of target size and the number of measurement points on the proposed method are evaluated through static target experiments, and the optimal target design criteria are proposed to improve our method’s performance in mitigating optical turbulence errors. Overall, this is a well written manuscript and the results are clear and compelling. Please consider the following comments:

(1)    The authors need to add the novel contribution and some significant results in the abstract of this manuscript.

(2)    The format of letter symbols in the full paper needs to be unified, such as italics or orthography. Specially, unify the font size in Figure 5.

(3)    The proposed method achieves an RMSE correction rate of 69% after mitigating the optical-turbulence error. It seems that the effect is not very high. How to apply it to practical projects.

(4)    The format of references needs to be unified.

(5)    The proposed method seems to be just a combination of several traditional methods, without seeing the innovation of the method.

Reviewer 2 Report

This paper need many sigificant changes before consider for publication in many aspects including structure, organization , writing and scientific contents. The authors have also should add more results and compare with state of art approaches. 

Reviewer 3 Report

The topic of manuscript is interesting due to constant development of computer vision-based displacement measurement techniques and its application for structural health monitoring.

The content and layout of the manuscript are correct. The quality of the paper can be improved if authors choose to take into account the following remarks:

- the review of the literature and the current state of knowledge should be extended. I suggest you read the review papers focused on this issue, of which you can find a lot;

- the measurement system and sensors should be described in more detail, basic technical data is welcome;

- it is worth presenting a more in-depth discussion of the results, especially regarding power spectral density before and after mitigation of optical turbulence errors, filtering out the frequency components after mitigation requires a comment and justification that no significant information was lost in the signal,

- the conclusions section can be extended.

Round 2

Reviewer 1 Report

It's OK for me. 

Reviewer 2 Report

No more comments